# Effect of Etching Depth on Threshold Characteristics of GaSb-Based Middle Infrared Photonic-Crystal Surface-Emitting Lasers

**DOI:** 10.3390/mi10030188

**Published:** 2019-03-14

**Authors:** Zong-Lin Li, Shen-Chieh Lin, Gray Lin, Hui-Wen Cheng, Kien-Wen Sun, Chien-Ping Lee

**Affiliations:** 1Department of Electronics Engineering and Institute of Electronics, National Chiao Tung University, Hsinchu City 30010, Taiwan; martin323261@gmail.com (Z.-L.L.); z810481@gmail.com (S.-C.L.); hwcheng2011@gmail.com (H.-W.C.); cplee@mail.nctu.edu.tw (C.-P.L.); 2Center for Nano Science and Technology, National Chiao Tung University, Hsinchu City 30010, Taiwan; 3Department of Applied Chemistry, National Chiao Tung University, Hsinchu City 30010, Taiwan; kwsun@mail.nctu.edu.tw

**Keywords:** photonic crystals, surface-emitting lasers, middle infrared lasers, GaSb-based lasers

## Abstract

We study the effect of etching depth on the threshold characteristics of GaSb-based middle infrared (Mid-IR) photonic-crystal surface-emitting lasers (PCSELs) with different lattice periods. The below-threshold emission spectra are measured to identify the bandgap as well as band-edge modes. Moreover, the bandgap separation widens with increasing etching depth as a result of enhanced diffraction feedback coupling. However, the coupling is nearly independent of lattice period. The relationship between threshold gain and Bragg detuning is also experimentally determined for PCSELs and is similar to that calculated theoretically for one-dimensional distributed feedback lasers.

## 1. Introduction

Semiconductor lasers emitting in the middle infrared (Mid-IR) range have promising applications in gas sensing, environmental monitoring, and military explosives detection [1,2,3]. Gallium antimonide (GaSb) and GaSb-related semiconductors are ideally suited for such light emitters because of the narrow energy bandgap as well as the Type-I quantum well (QW) heterostructure [2]. It is worth mentioning that Mid-IR sensors based on tunable diode laser absorption spectroscopy (TDLAS) render the lasers in single spectral mode with a narrow linewidth. 

In recent years, photonic-crystal surface-emitting lasers (PCSELs) have attracted a lot of attention because of their narrow spectral linewidth, high output power, and small beam divergence angle [4,5,6]. By properly designing two-dimensional (2D) photonic crystals (PhC) that satisfy a specific Bragg condition, light waves from gain media can couple with PhC, whereby a 2D cavity mode is constructed to produce lasing emissions from the surface of the device. GaSb-based PCSELs are designed in connection to the abovementioned TDLAS sensors; however, only optically pumped devices are successfully demonstrated [5,7], while electrically pumped ones are still in development [8].

The design and fabrication of PCSELs without a complicated technology of regrowth or fusion bonding is preferred, but weak diffraction feedback coupling between the PhC layer and QW active region is resulted because the optical mode is pushed away from the PhC layer by low-index ambient and etched holes. To compensate for weak coupling, the etching depth of PhC holes is increased to reduce the threshold pumping density. Prior works involved two or three etching depths [9,10]; no systematic study on depth effect has been conducted.

In this work, we report the effect of etching depth on the threshold characteristics of PCSELs with different lattice periods. GaSb-based gain media are selected as they are easily pumped with a low threshold pumping density [5,7]. The below-threshold emission spectra are also measured to identify the bandgap as well as the band-edge modes. The bandgap separation is analyzed with respect to etching depth and lattice period. Finally, the relationship between threshold gain and Bragg detuning is experimentally determined and discussed.

## 2. Materials and Methods 

The sample in this work was grown on (001) n-type GaSb substrates using a Veeco GEN II molecular beam epitaxy system. It consisted of, from the bottom upward, a 200-nm GaSb buffer layer, a 2000-nm Al_0.85_Ga_0.15_As_0.07_Sb_0.93_ bottom-cladding layer, a 150-nm Al_0.3_Ga_0.7_As_0.02_Sb_0.98_ separate confinement layer (SCL), two layers of 10-nm In_0.35_Ga_0.65_As_0.14_Sb_0.86_ quantum well (QW) spaced by 20-nm Al_0.3_Ga_0.7_As_0.02_Sb_0.98_ layer, a 200-nm Al_0.3_Ga_0.7_As_0.02_Sb_0.98_ SCL, a 200-nm top-cladding layer Al_0.5_Ga_0.5_As_0.04_Sb_0.96_, and a 200-nm GaSb capping layer. The entire structure is shown in Figure 1a.

A 150-nm-thick Si_3_N_4_ layer was first deposited as a hard mask by plasma-enhanced chemical vapor deposition. The PhC structure was then fabricated using electron-beam lithography and an inductively coupled plasma reactive ion etching system. Finally, the hard mask was removed before optical pumping. The pattern of PhC was designed in a square lattice with circularly shaped air holes. An array of 5-by-5 PhC regions (five rows indexed 1 to 5 by five columns indexed A to E) was patterned in a small wafer and cut into five pieces for different etching depth. The lattice periods varied from 620 nm to 640 nm in steps of 5 nm and were labeled 1 to 5 with increasing period. PhC holes with different depths of 150, 200, 250, 300, and 350 nm were dry-etched and labeled A to E with increasing depth. Therefore, a total of 25 PCSEL devices were prepared for the measurement, i.e., devices A1 to A5, B1 to B5, C1 to C5, D1 to D5, and E1 to E5. The filling factor, defined as the ratio of hole area within unit lattice, was 0.1 and each device area was 300 × 300 µm^2^. Figure 1b is the cross-sectional scanning electron microscope (SEM) image of a PCSEL.

The PCSEL devices were fixed on a copper plate. The temperature of the plate was controlled at 20 °C using a temperature controller (LDT-5545B, ILX Lightwave Co., Bozeman, MT, USA). The devices were pumped by a 1064-nm pulsed fiber laser (MOPA-PS/CDRH 12V, Multiwave Photonics S.A., Maia, Portugal) with a pulse width of 100 ns and a repetition rate of 5 kHz. It should be noted that a CaF_2_ lens (LA5370, Thorlabs Inc., Newton, NJ, USA,) was used to focus the pumping laser beam as well as to collect the Mid-IR emissions normally to the device surface. The illuminated spot was 200 μm in diameter, which approximately filled the entire area of the device. A longpass filter was used to block the 1064 nm pumping light. Finally, another CaF_2_ lens was used to focus the Mid-IR light into a monochromator (iHR 320, Horiba Jobin Yvon Inc., USA) with a thermoelectrically cooled InGaAsSb detector (IGA2.2-010-TC, Electro-Optical Systems Inc., Phoenixville, PA, USA). The resolution of the monochromator was 0.1 nm. Figure 2 shows the setup of the optical pumping system.

## 3. Results and Discussions

The 25 devices were optically pumped to exhibit surface lasing emissions. The lasing spectra were acquired at pumping power of 1.2 times the threshold. Figure 3a shows the normalized spectra for devices A4, B4, C4, D4, and E4 (with a fixed lattice period of 635 nm). The spectral full width at half maximum (FWHM) ranges from 0.2 nm to 0.3 nm, and the corresponding quality factor (Q-factor) distributes from a little more than 7000 to over 10,000. The peak lasing wavelengths of the 25 devices are plotted against individual etching depths in Figure 3b. The linear dependence is fitted for five different lattice periods with an almost equal slope of −0.05, i.e., a wavelength shift of −5 nm for every 100-nm increase in etching depth. At a fixed etching depth, the wavelength shift is about +3.14 nm per nm increase in lattice period, which is consistent with InP- and GaAs-based PCSELs [10,11].

The RT photo-luminescence (PL) was peaked around 2172 nm with spectral width at 95% intensity of about 50 nm (not shown). The 635-nm-period devices (A4 to E4), which lased around the gain peak, exhibited the lowest pumping threshold among the different devices. Figure 4a shows the curves of light-in versus light-out (*L-L*) for devices A4 to E4. The slope efficiency is not discussed because there may be large deviations between measurements. The plot of threshold power density versus etching depth is shown in Figure 4b. The deeper the PhC holes are etched, the more enhanced feedback couplings between PhC layers and QW active region are achieved in which lower threshold gain is resulted. Therefore, the threshold power decreases rapidly with increasing depth and then levels off to become saturated. The dependence of threshold power density on etching depth can be fitted by an exponential function with an offset of 225. The offset is attributed to minimum achievable threshold gain, which will be discussed later.

Figure 5a shows the etching depth dependence of threshold power density for 620- and 630-nm-period devices (A1 to E1 and A3 to E3). They are fitted by exponential functions with the offsets of 315 and 285 for 620- and 630-nm-period devices, respectively. Based on our previous works [9,10,11], devices with less gain-cavity offset and deeper etching depth exhibited a lower threshold pumping density. Therefore, we excluded devices D1 and E3 from fitting according to the above criteria. The threshold power densities for device A2 to E2 and A5 to E5 are also plotted again etching depths in Figure 5b. However, no exponential dependence is observed for the 625- and 640-nm-period devices. Note that the gain-cavity offsets fell between 620- and 635-nm-period devices, but higher pumping thresholds were observed for devices B2, B5, C2, C5, and E2. We attributed the causes to growth imperfection, process variation, and/or a misalignment in focusing. As a result, the threshold pumping densities of the 625- and 640-nm-period devices are not analyzed afterwards.

Because feedback coupling between PhC layers and QW active region is enhanced with increasing etching depth, the normalized frequency difference between band-edge modes is expected to widen. Based on the measurement setup in Figure 2, we collected below-threshold emissions by optical lens with numerical aperture of about 0.3. Since angular information cannot be resolved, only band-edge modes B and C are identified by their intensity contrast with mode bandgap. Figure 6a shows the above-mentioned spectra for 635-nm-period devices (A4 to E4). The longer-wavelength and shorter-wavelength peaks in the immediate vicinity of intensity bandgap are ascribed to band-edge modes B and C, respectively. Moreover, the bandgap between modes B and C increases from 10.5 nm to 18.5 nm with increasing etching depth. It is an indication of enhanced feedback coupling and, as a result, lower threshold gain is expected and consistent with our observation of lower threshold pumping power. Figure 6b shows the below-threshold spectra for 350-nm-deep but varying period devices (E1 to E5). The bandgap separation is almost equal for five devices with fixed etching depth but different lattice period. The feedback coupling is nearly independent of lattice period.

The identified band-edge modes are normalized in frequency by dividing lattice periods by vacuum wavelengths and plot against etching depths as shown in Figure 7a. Only 630- and 635-nm-period devices are plotted for clarity. Figure 7b shows their normalized lasing and Bragg frequencies versus etching depths. The lasing peaks are originated from band-edge mode B, which is the same as revealed in our previous work [6,11]. Regarding the Bragg frequencies, these are determined as the center of band-edge modes B and C. Therefore, the normalized lasing frequency is detuned from the Bragg condition by a detuning parameter (δ < 0), whose magnitude increases with increasing etching depth.

Assuming that the logarithmic relationship between threshold gain (*gth*) and current density holds for optical pumping, its inverse relationship,
(1)$$P_{th}\left(g_{th}\right)=P_{0}exp\left(\frac{g_{th}-g_{0}}{g_{0}}\right),$$
is modeled by two parameters of reference gain (*g*_0_) and associated pumping density (*P*_0_). Let the offsets in Figure 4b and Figure 5a correspond to the same minimum achievable modal gain (say *g*_0_ = 10 cm^−1^) for PCSEL devices. Therefore, the net modal gain can be extracted, while the detuning is extracted from Figure 6b. Figure 8 show the threshold modal gain as a function of normalized frequency detuning. The vertical axis is relative to the reference gain. As the etching depth increases, the feedback coupling is enhanced to result in larger detuning but a lower threshold gain. The gain-detuning relationship is experimentally determined to be exponential-like and similar to theoretical predictions for the first-order modes of one-dimensional (1D) distributed feedback (DFB) lasers [12].

## 4. Conclusions

In this paper, we have systematically studied the device characteristics of GaSb-based Mid-IR PCSELs with respect to etching depth and lattice period. In terms of lasing wavelength, the wavelength shift is about +31.4 nm for every 10-nm increase in period and −5 nm for 100-nm increase in depth. Measurement of below-threshold emission spectra identifies the bandgap as well as band-edge modes. The bandgap separation, which is a function of feedback coupling, increases with increasing depth but is independent of the lattice period. The criteria are set to select devices for threshold gain analysis. With increasing depth, the threshold pumping density decreases exponentially to a saturation level, which is assigned to the minimum device modal gain. The relative threshold gain is then plotted as a function of normalized frequency detuning. The gain-detuning relationship of PCSELs is similar to that of 1D DFB lasers.

## Figures and Tables

**Figure 1 micromachines-10-00188-f001:**
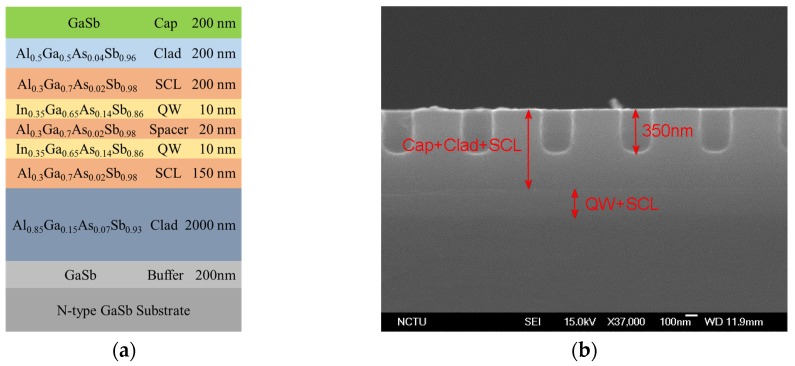
(**a**) The schematic diagram of sample structure. (**b**) The cross-sectional SEM images of a PCSEL.

**Figure 2 micromachines-10-00188-f002:**
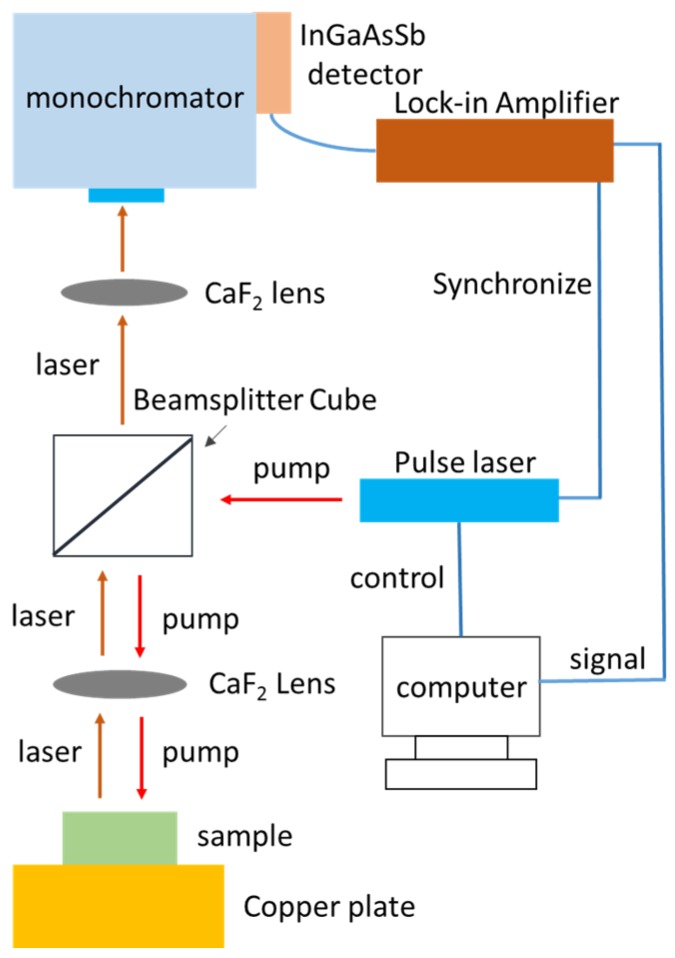
The measurement setup of optical pumping system.

**Figure 3 micromachines-10-00188-f003:**
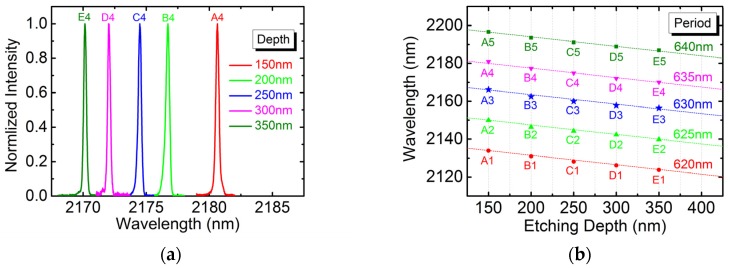
(**a**) The normalized lasing spectrum for devices with fixed lattice period of 635 nm. (**b**) The peak lasing wavelength versus etching depth for 25 devices.

**Figure 4 micromachines-10-00188-f004:**
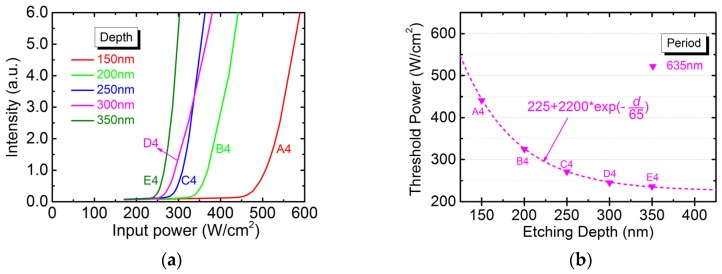
(**a**) The light-in versus light-out (*L-L*) curves and (**b**) the dependence of threshold power density on the etching depth for 635-nm-period devices.

**Figure 5 micromachines-10-00188-f005:**
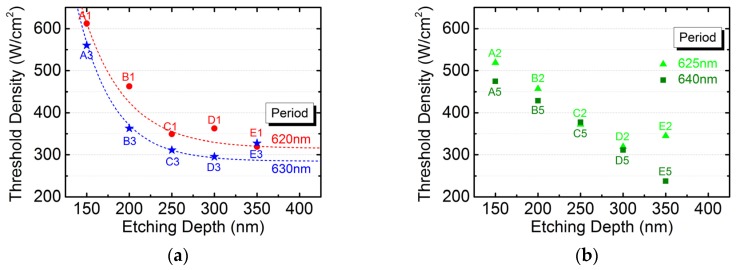
The dependence of threshold power density on the etching depth for (**a**) 620- and 630-nm-period devices (A1 to E1 and A3 to E3) as well as (**b**) 625- and 640-nm-period devices (A2 to E2 and A5 to E5).

**Figure 6 micromachines-10-00188-f006:**
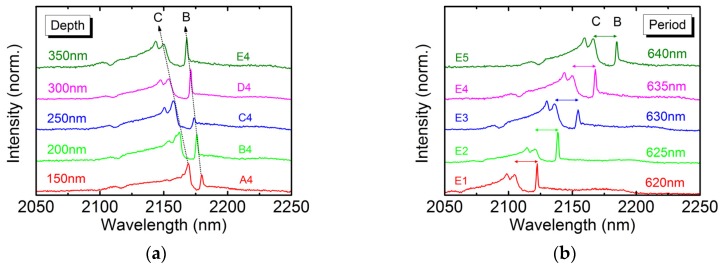
The below-threshold emission spectra for devices with (**a**) 635-nm-period but varying etching depth (A4 to E4) and (**b**) 350-nm-deep but varying period (E1 to E5).

**Figure 7 micromachines-10-00188-f007:**
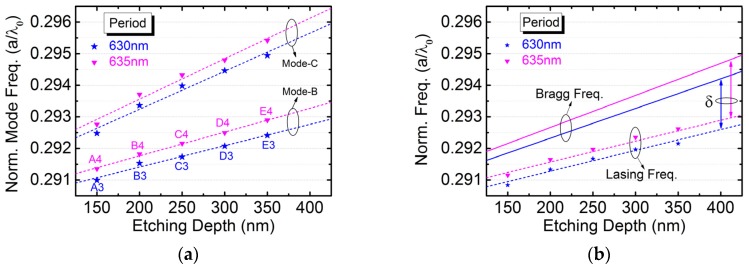
(**a**) The normalized band-edge mode frequency as well as (**b**) the normalized lasing and Bragg frequency is plotted against etching depth for 630- and 635-nm-period devices.

**Figure 8 micromachines-10-00188-f008:**
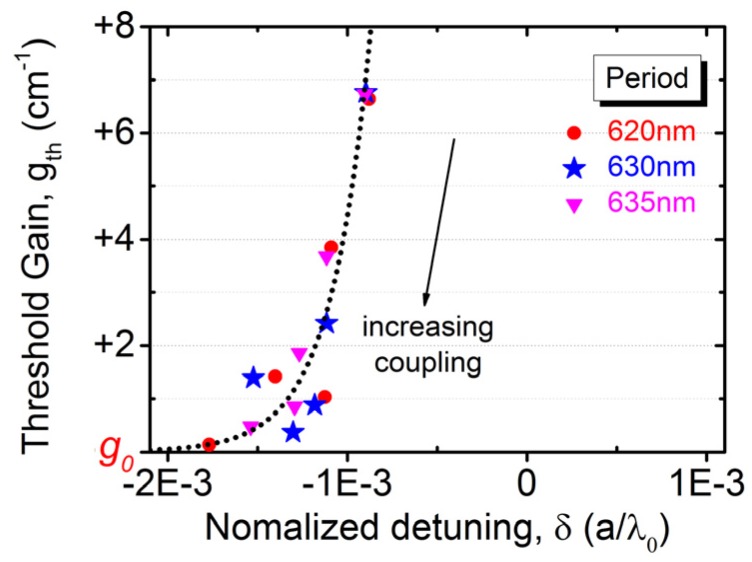
The threshold modal gain is plotted as a function of normalized frequency detuning.

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
