# Peer review of "Effect of Etching Depth on Threshold Characteristics of GaSb-Based Middle Infrared Photonic-Crystal Surface-Emitting Lasers"

_micromachines, 2019, doi:10.3390/mi10030188_

Round 1

Reviewer 1 Report

This manuscript presents a systematical study the effect of etching depth on the threshold characteristics of GaSb-based middle infrared (Mid-IR) photonic-crystal surface-emitting lasers (PCSELs) with different lattice periods. In general, the manuscript is well written and the work is meaningful. I think it can be accept after minor revision.

Here are some comments for authors to consider:

1) Figure 1: There is no photonic crystals structure shown in it. A cross-sectional (or 3D) schematic diagram or a SEM  photograph with periodical PhC holes should be used to show the device structure.

2) Line 53-69: Please provide more details of the fabricated device. Has the  hard mask Si3N4 layer been removed? is there any other layer coated on the device surface? a

3)Line 70-71: Please provide the testing temperature. 

4) Figure 5b and line 108-112: The explanation for Figure 5b is too rough, and the manufacturing issues like imperfect fabrication and careless alignment can be solved. If possible, please provide results of devices without these problems. 

5) Figure 8 (Line153): Please check the caption of Figure 8. It should be "the threshold modal gain as a function of normalized frequency detuning".?

Author Response

Attached please find our response in PDF.

Reviewer 2 Report

This manuscript investigates the effect of etching depth on GaSb-based Mid-IR PCSELs with different periods. The topic is of interest and the paper is easy to follow. Some minor comments are suggested to improve the quality of the paper as below:

1) Line 81: "The 25 devices..." why select these 25 devices to test? Any simulation to determine why these 25 devices are the critical values to investigate?

2) Figure 5 and Line 106-112: From figure 5 (a) it can be seen that D1 and E3 in figure 5a did not follow the trend. Also Figure 5b follows a different trend. A more detail explanation or more experiments or simulation will be needed to tell why it happens this way and how to design this effect for density control based on these findings?

3) Line 133-134, why only 630nm and 635 nm were needed to be plotted?

4) More details in conclusion are recommended.

Author Response

(The authors gave the same response as above.)
